# How Microbial Food Web Interactions Shape the Arctic Ocean Bacterial Community Revealed by Size Fractionation Experiments

**DOI:** 10.3390/microorganisms9112378

**Published:** 2021-11-17

**Authors:** Oliver Müller, Lena Seuthe, Bernadette Pree, Gunnar Bratbak, Aud Larsen, Maria Lund Paulsen

**Affiliations:** 1Department of Biological Sciences, University of Bergen, 5006 Bergen, Norway; Bernadette.Pree@uib.no (B.P.); Gunnar.Bratbak@uib.no (G.B.); 2Department of Arctic and Marine Biology, UiT—The Arctic University of Norway, 9037 Tromsø, Norway; lena.seuthe@uit.no; 3Molecular Ecology Group, NORCE, 5008 Bergen, Norway; aula@norceresearch.no; 4Arctic Research Center, Department of Ecoscience, Aarhus University, 8600 Silkeborg, Denmark; mlp@bios.au.dk

**Keywords:** microbial food web, experimental manipulations, trophic interactions, seasonal changes, Arctic Ocean, bacterial community structure, microbial resilience, phytoplankton–bacteria association

## Abstract

In the Arctic, seasonal changes are substantial, and as a result, the marine bacterial community composition and functions differ greatly between the dark winter and light-intensive summer. While light availability is, overall, the external driver of the seasonal changes, several internal biological interactions structure the bacterial community during shorter timescales. These include specific phytoplankton–bacteria associations, viral infections and other top-down controls. Here, we uncover these microbial interactions and their effects on the bacterial community composition during a full annual cycle by manipulating the microbial food web using size fractionation. The most profound community changes were detected during the spring, with ‘mutualistic phytoplankton’—Gammaproteobacteria interactions dominating in the pre-bloom phase and ‘substrate-dependent phytoplankton’—Flavobacteria interactions during blooming conditions. Bacterivores had an overall limited effect on the bacterial community composition most of the year. However, in the late summer, grazing was the main factor shaping the community composition and transferring carbon to higher trophic levels. Identifying these small-scale interactions improves our understanding of the Arctic marine microbial food web and its dynamics.

## 1. Introduction

Marine microorganisms play a major role in global nutrient cycles. Understanding the factors that control their distribution and activity is fundamental to improving our insight into ecosystem functions. Over the last decades, the concept of a microbial loop [1] amended by a viral shunt [2] was established as the “road map” to describe how dissolved organic material originating from a range of biological processes (e.g., cell lysis, leakage, exudation, excretion, sloppy feeding, etc.) is returned to higher trophic levels via bacteria and phagotrophic grazers. Recent focus has been on unraveling the various factors that affect connections at the base of the food web and its upward cascading effects on larger key predators, such as micro- and mesozooplankton [3,4]. Only few studies have considered the downward cascading effects, i.e., how larger organisms may shape the base of the food web by grazing or carbon production.

At high latitudes, where the light availability changes significantly over the course of the year, the feedback between phototrophic growth and the heterotrophic utilization of a generated biomass determines the fate of the carbon through the microbial food web [5,6]. Thus, seasonality has a strong impact on driving microbial community changes [7]. While small phytoplankton cells (<20 µm) dominate during the winter and early spring, the retreating ice edge during the spring will give rise to large-celled phytoplankton, such as diatoms [6,8,9,10]. This period, with increased phytoplankton activity and productivity, is relatively short in the Arctic [10,11]. These bloom events are followed by shifts towards phytoplankton communities that are based on regenerated production. These distinct seasonal phytoplankton shifts have multiple effects on the entire microbial food web [12].

To gain a better and predictive understanding on how organic matter is transported from autotrophs to higher trophic levels under different ecological states of the ecosystem, it is necessary to study how the different components of the microbial food web interact. Previous studies have shown that external factors such as the availability of mineral nutrients and dissolved organic carbon (DOC) and hydrographical conditions affect the community structure, which, in turn, is important for the overall production in and export from the system [3,13,14]. In a simplified “minimum” microbial food web model, Thingstad et al. [15] described how bacteria, autotrophic flagellates and diatoms (i.e., the osmotrophs) compete for mineral nutrients (N and P) (Figure 1a). Bacteria and diatoms are, in addition, controlled by the supply of DOC and Si, respectively. The predators, heterotrophic flagellates, microzooplankton (e.g., ciliates) and mesozooplankton (i.e., the phagotrophs) each exert a top-down control on smaller forms and constitute a grazing food chain.

Environmental changes occurring in the Arctic will affect the organisms at the base of the food web with cascading effects on the ecosystem’s functions. To a large extent, these changes are related to the increased Atlantic water inflow [16,17], of which the West Spitsbergen Current is a major source. This large body of warm and saline Atlantic Water causes increased sea ice melt and affects the water column stability and vertical mixing depth [18,19]. In turn, this affects the nutrient content and, hence, competition for nutrients, as well as light conditions for organisms inhabiting the photic zone, all of which have consequences for the composition and activity of phytoplankton communities and associated bacterial communities.

In the present study, we aimed at identifying how the microbial food web is controlled under changing seasonal conditions and in different physicochemical environments at the inflow to the Arctic Ocean (Figure 1b). We hypothesize that the bacterial growth is highest during the productive time of year when nutrient concentrations are high and that phytoplankton abundance and composition determine the bacterial community structure. During the post-bloom phases when regenerated production dominates, we expect the grazing pressure to be the highest and hypothesize top-down control to be an important driver of the bacterial community structure. We tested our hypotheses by manipulating predator communities by size fractionation in grazer exclusion experiments conducted at five different times of the year (Figure 1a,b). We further hypothesized two possible outcomes: firstly, that different size fractions will result in different bacterial communities with activity and composition changes over time (I), or secondly, that bacterial communities in all fractions would remain similar throughout the experiment (II). The underlying causes for both scenarios will likely be due to substrate specificity, co-association with other microorganisms or grazing. By combining abundance measurements with diversity analyses, we obtained insights into how the bacterial community is controlled, how the different size groups interact and how nutrients flow through the system. This information helps to interpret shifts between bottom-up and top-down control states in the microbial food web.

## 2. Materials and Methods

### 2.1. Study Site

The experiments were conducted once during each of the five research cruises NW of Svalbard in 2014 (Figure 1b). The cruises covered different times of the year and took place in January (06.01–15.01), March (05.03–10.03), May (15.05–02.06), August (07.08–18.08) and November (03.11–10.11). During May, August and November, transects crossed the West Spitsbergen Current (WSC), the core of the Atlantic Water (AW) inflow, at 79° N and 79.4° N. In January and March, the cruises followed the WSC further north, where its southern branch into the Arctic Ocean was investigated at 80.5° E to 82.6° N. Water samples for the experiments were collected at 20 m, coinciding with the Deep Chlorophyll Maximum (DCM) in May and August.

### 2.2. Experimental Design

Grazer exclusion experiments allowing for growth rate measurements of different microbial groups in the absence of their grazers were performed [20,21,22,23]. Surface water (20 m) samples were gently reverse-filtrated, using different mesh size filters (0.8, 10 and 90 μm) to retain organisms of different size fractions. The smallest size group (<0.8 µm) represented the bacterial fraction (the term bacteria here referring to prokaryotic organisms, as it also included archaea) without grazers and producers larger than 0.8 µm. The <10-µm fraction included bacteria and small autotrophic organisms (cyanobacteria, picophytoplankton and nanophytoplankton), as well as small heterotrophic grazers such as heterotrophic nanoflagellates (HNF). The largest fraction (<90 µm) comprised larger phytoplankton, ciliates and other microzooplankton, in addition to all the smaller forms (Figure 1a). Thus, bacteria were either released of any grazing pressure (<0.8-µm fraction); exposed to grazing by HNF (<10-µm fraction) or reduced grazing (<90-µm fraction), as microzooplankton grazing HNF may lower the grazing on bacteria. The bacterial growth and activity may, in all three fractions, also be limited by the availability of mineral nutrients and organic carbon and by viral infections. After the filtration, water from every size fraction was transferred into 3.9-L transparent polycarbonate bottles (Nalgene^®^) using silicone tubing and staggered filling. The experiment bottles were incubated at close to in situ temperatures and light conditions. During May and August, the incubations were placed in plexiglass tanks on deck with a continuous seawater flowthrough (May: 1.7 ± 1.6 °C and August: 1 ± 0.8 °C), and nylon wrapping around each bottle reduced the PAR to 30% of the surface irradiance. The incubations in January, March and November were placed in a dark cooling room at 2 °C, and only in March, artificial light (5-μmol photons m^−2^ s^−1^) was supplied to simulate the in situ light cycle (16-h darkness and 8-h light). The experiments were incubated for 5–10 days, where the samples for microbial abundance measurements were taken daily and nutrient samples every second day. The samples for microbial community composition were obtained at the start and end of the experiments in March, May, August and November (samples were also taken from the experiments in January; however, the quality was insufficient to be included in this study).

### 2.3. Environmental Parameters (Nutrients, DOC and Light)

For the analysis of the nutrients, seawater samples were directly poured into 30-mL acid-washed HDPE bottles and stored at −20 °C. Silicic acid (H_4_SiO_4_), nitrite and nitrate (NO_2_^−^ + NO_3_^−^) and phosphate (PO_4_^3−^) were measured on a Smartchem 200 (AMS Alliance, Rome, Italy) autoanalyzer following the procedures as outlined in References [24,25,26], respectively. Samples for the analysis of dissolved organic carbon (DOC) were collected in 60-mL acid-washed HDPE bottles and stored at −20 °C. The concentrations were determined by high-temperature combustion (720 °C) using a Shimadzu TOC-V CPH-TN carbon analyzer. The samples were acidified to pH 2 using concentrated hydrochloric acid. The instrument was calibrated using acetanilide [27], and carbon was determined by the community deep-sea reference (Hansell Laboratory, Miami, FL, USA). The light estimates used to describe the seasonal variations in light were calculated based on daily averages at 79° N over an annual cycle.

### 2.4. Chlorophyll a (Chl a)

Between 0.25 and 1 L of water were collected for determination of chlorophyll *a* (Chl *a*) concentrations. Water samples were filtered in triplicate onto non-combusted 25-mm Whatman GF/F filters using vacuum pumps and low pressure (~30 kPa). Following filtration, Chl *a* was extracted from the filters in 5-mL 96% methanol for 12–24 h and analyzed on a Turner Design AU10 Fluorometer calibrated against a Chl *a* standard.

### 2.5. Microbial Abundance Measurements Using Flow Cytometry

The abundance of microbial organisms, including virus, bacteria, heterotrophic nanoflagellates (HNF) and pico- and nanosized phytoplankton, were determined using an Attune^®^ Acoustic Focusing Flow cytometer (Applied Biosystems by Thermo Fisher Scientific, Waltham, MA, USA)) with a syringe-based fluidic system and a 20-mW 488-nm (blue) laser. Samples were prepared in triplicate by fixing 2 mL of water with glutaraldehyde at 4 °C for a minimum of 2 h, flash-frozen in liquid nitrogen and stored at −80 °C. Prior to the analysis of the virus and bacteria, the samples were first thawed, diluted 10 times with 0.2-μm-filtered TE buffer (Tris 10 mM and EDTA 1 mM, pH 8), stained with a green fluorescent nucleic acid dye (SYBR Green I; Molecular Probes, Eugene, OR, USA) and then incubated for 10 min at 80 °C in a water bath [28]. The stained samples were counted at a low flow rate of 25 µL min^−1^ and a minimum volume of 100 µL, and different groups were discriminated on a biparametric plot of green fluorescence (BL1) vs. red fluorescence (BL3). This allowed distinguishing the virus particles, low nuclear acid (LNA) and high nuclear acid (HNA) bacteria. For the HNF analysis, the samples were stained with SYBR Green I for 2 h in the dark, and subsequently, 1 to 2 mL were measured at a flow rate of 500 µL min^−1^ following the protocol of Zubkov et al. [29]. Autotrophic pico- and nanosized plankton were counted after thawing, and the various groups were discriminated based on their red fluorescence (BL3) vs. orange fluorescence (BL2) [30]. Changes in the cell (and virus) abundances were used as a measure of the activity/net growth.

### 2.6. Light Microscopy

To enumerate organisms that are too large to be detected using flow cytometry, a microscopic analysis was performed for samples from the <90-µm fractions. One hundred and ninety milliliters of seawater were poured into 200-mL brown glass bottles and were preserved with Lugol’s acid solution (final conc. 2%) and gently mixed. Samples were stored at 4 °C until further processing. Organisms were identified to the lowest taxonomic level possible using an inverted light microscope and phytoplankton and protists quantified (abundance in cells m^−3^).

### 2.7. Bacterial Production (BP)

In addition to changes in the cell abundance, bacterial activity was also measured as the bacterial production, estimated from the incorporation of 3H-leucine according to Smith & Azam [31]. Four replicates of 1.5 mL from the water samples were transferred to 2-mL plastic vials. Eighty microliters of 100% trichloroacetic acid (TCA) were immediately added to one sample and served as a control. Samples were incubated with 25-nM 3H-leucine (final concentrations) for 2 h at the in situ temperature, and the incubations were stopped by an addition of 80 µL of 100% TCA. Samples were then centrifuged for 10 min at 14,800 rpm and subsequently washed with 5% TCA twice. For analysis, 5 mL of scintillation liquid (Ultima Gold) was added, and the radioactivity was counted on a Perkin Elmer Liquid Scintillation Analyzer Tri-Carb 2800TR. The measured leucine incorporation was converted to g carbon incorporated per L per hour (according to Simon & Azam [32]) using the specific activity of the isotope and the constant 1797 (grams of protein produced per mole of incorporated leucin) and 0.86 (the weight ratio (g:g) of total C:protein in bacteria).

### 2.8. Nucleic Acid Extraction and Amplicon Library Preparation

Molecular samples were collected by filtering 2 L onto 0.22-µm Sterivex filters. DNA and RNA were coextracted from the Sterivex filters using the AllPrep DNA/RNA Mini Kit (Qiagen, Hilden, Germany). For this study, only RNA was used for the molecular analyses. The RNA was treated with the DNA-free DNA Removal kit (Invitrogen, Waltham, MA, USA), and reverse-transcribed using the SuperScript III First-Strand Synthesis System for RT-PCR (Invitrogen). Subsequent amplification of the cDNA was done using a two-step nested PCR approach with primers (519F and 806R) targeting both the archaeal and the bacterial 16S rRNA gene V4 hypervariable regions. Libraries were sequenced at the Norwegian Sequencing Centre (Oslo, Norway) using their MiSeq technology (MiSeq Reagent Kit v2, Illumina, San Diego, CA, USA).

### 2.9. Sequence Analysis

The retrieved paired-end sequence data were processed using different bioinformatic tools incorporated on a qiime-processing platform [33], as described by Wilson et al. [7]. In short, the FASTQ files were quality end-trimmed and merged, and the OTUs (operational taxonomic units) were selected at a sequence similarity threshold of 97%. Taxonomy was assigned using the Greengenes reference database [34]. All chloroplast reads, including the eukaryotic plastidial and prokaryotic cyanobacterial sequences, were filtered out, and the eukaryotic plastidial sequences were taxonomically reassigned using the Phytoref database of the plastidial 16S rRNA gene of photosynthetic eukaryotes [35].

### 2.10. Functional Prediction with Tax4Fun

To predict the functional potential of the prokaryotic community, the software package Tax4Fun [36] was applied in R. The application is designed to match 16S rRNA gene datasets to organisms represented in rRNA databases, such as SILVA, and predict the metabolic potential based on the abundance of matched OTUs. It uses only OTUs that could be mapped to KEGG (Kyoto Encyclopedia of Genes and Genomes) [37] organisms. The KO (KEGG Orthology) relative abundance in the samples was analyzed using MEGAN5 [38] and the functional pathways identified based on KEGG classification.

### 2.11. Statistical Analysis

An analysis of the OTU tables, including calculations of the alpha diversity indices, Bray–Curtis dissimilarities between samples and multidimensional scaling analysis, were carried out using the program primer-e version 6 (Plymouth, UK). Calculations for the Pearson correlation coefficient (Pearson’s *r*) were carried out using GraphPad Prism v 9.01 for Windows (GraphPad Software, San Diego, CA, USA). Pearson’s *r* linear correlations were carried out to investigate the significance of the correlations between the bacterial production data and percentage of high nucleic acid bacteria. We calculated the alpha diversity changes, including richness and evenness, expressed as chao1, of the bacterial communities based on 16S rRNA gene sequencing from samples taken at the start and end time points of the experiments. The alpha diversity changes were reflected in the composition of the bacterial community, and similarities between communities were investigated and visualized through multidimensional scaling (MDS) plots.

## 3. Results

The five experiments represent different seasons of the Arctic ecosystem, i.e., the polar night (January), the onset of light (March), spring (May), summer (August) and the return to the polar night (November) (Figure 2a). The environmental characteristics, including light; nutrients (NO_x_, Si and PO_4_); the amount of total dissolved organic carbon (DOC) and chlorophyll *a* (Chl *a*) concentration (Figure 2a and Figure A1), were thus different at the start of each experiment, with the nutrients being at their minimum in August while DOC and total Chl *a* peaked in May.

### 3.1. Seasonal Changes

The initial abundance of the microorganisms varied for all functional groups, with lowest concentrations in the late winter months (January and March) and highest in May and August (Figure 2b and Figure A1). The size fractionation set-up was considered successful, as the initial abundances of the bacteria and virus were comparable in all size fractions and larger organisms were largely absent in the <0.8-µm fraction. The initial abundance of the larger organisms was initially equal to or higher in the <90-µm than in the <10-µm fraction. Most groups showed the highest net growth in May in the largest size fraction (<90 µm).

The virus abundances were similar for all fractions and peaked during the August experiment with 3 × 10^7^ virus mL^−1^. The bacterial abundances varied between 1 × 10^5^ cells mL^−1^ in January and 1 × 10^6^ cells mL^−1^ in August. The bacterial net growth was highest in May in the <10-µm- and <90-µm-sized fractions, while, in August, it was negative in these fractions and slightly positive in the <0.8-µm fraction (Figure 2b). The initial abundance of heterotrophic nanoflagellates (HNF) was low in May (100 cells mL^−1^), but the net growth was highest in the <90-µm fraction, where they reached >2 × 10^3^ cells mL^−1^ within six days. Their overall abundance was highest in August (2–4 × 10^3^ cells mL^−1^), but no clear increase was observed during incubation.

For the autotrophic organisms, *Synechococcus*, pico- and nanophytoplankton, the highest abundances and net growth were observed in May and August. *Synechococcus* was detectable at all time points in the <10- and <90-µm fraction (~100 cells mL^−1^), but the overall highest abundance was measured at the beginning of the experiment in August with 1.0 × 10^4^ cells mL^−1^, followed by a continuous decline over the six days of incubation. Picoeukaryotes reached the overall highest abundances after six days of incubation in the <10- and <90-µm fractions in the May experiment, when the abundance increased from 5.0 × 10^3^ cells mL^−1^ to 1.0 × 10^4^ and 1.7 × 10^4^ cells mL^−1^, respectively. The abundances of picophytoplankton were still high with 1.0 × 10^4^ cells mL^−1^ when the August experiment was set up. A slightly positive net growth over the first four days up to 1.2 × 10^4^ cells mL^−1^ was followed by a decline to 7.5 × 10^3^ cells mL^−1^ at the end of the experiment. Nanophytoplankton displayed the highest net growth and peak abundance in the <90-µm fraction in May, when we detected an increase from 1.0 × 10^3^ to 1.0 × 10^4^ cells mL^−1^ during the experiment. The initial concentrations were similar in the <10-µm fraction but declined from 700 to 100 cells mL^−1^. In August, the nanophytoplankton abundances followed trends similar to that of picophytoplankton, with relatively high starting concentrations and a decline after six days (4.0 × 10^3^ to 3.0 × 10^3^ cells mL^−1^).

### 3.2. Bottom-Up and Top-Down Controls of Bacterial Activity

Both the bacterial production (BP) and abundance, indicators of bacterial activity, increased over time during the incubation experiments, and the overall highest values were measured in May and August (Figure 3). Independent of the size fractions, there was an increase in bacterial abundance (net-growth) over time in the January, March and May experiments. In August and November, the bacterial abundance increased in the <0.8-µm and the <10-µm fractions, respectively, while it decreased in the entire (August) or latter part (November) of the experiment in the other fractions. While community production was highest in the May and August experiments, bacterial production per cell was, in general, higher in the January and March experiments, where the overall cell abundances were lower than in May and August (Figure A2).

Different patterns of bacterial activity based on the BP, abundance and development over time could be recognized, as shown in Figure 3. In May, there was the same pattern independent of the size fraction, while, in January, March and August, the pattern in the <10-µm and <90-µm fractions was different from the <0.8-µm fraction, with a more pronounced increase in BP at the end of the incubations in January and March and decrease in bacterial abundance in August. The overall highest activities were measured in August, with high (2 to 3-µg C L^−1^ d^−1^ at day 6) bacterial activity in all three size fractions. While there was also a similar high activity in the January and March experiments, there was a clear difference depending on the size fractions, with the lowest activity (0.1–0.4-µg C L^−1^ d^−1^ at day 6) in the <0.8-µm and highest (1–3-µg C L^−1^ d^−1^ at day 6) in the <90-µm fraction.

All experiments covering seasonal differences and size fractions showed a trend towards a coupling of high bacterial productivity and proportion of high nuclear acid (HNA) bacteria (Figure 4). Data points with high bacterial production and a high HNA to total bacteria ratio were predominantly from the latter days of the experiments.

### 3.3. Drivers of Bacterial Community Composition Changes

The bacterial community composition, based on 16S rRNA gene sequencing from the start (T_0_) and end (T_end_), showed lower diversity, expressed as chao1, in March, May and August compared to November (Figure 5). We observed a decrease in diversity between T_0_ and T_end_ in the <0.8-µm fraction for all four experiments and, in March and May, also in the other two size fractions. The lowest diversity in any of the samples was measured at the end of the May experiment in the <90-µm fraction. In August and November, the diversity increased in the two larger fractions between T_0_ and T_end_.

The alpha diversity changes are reflected in the composition of the bacterial community, and similarities between the communities were investigated and visualized through a multi-dimensional scaling (MDS) plot (Figure 5b). The MDS plot shows that there was a seasonal difference in the initial community compositions and that the community composition changed during incubation in all cases. The effect of size fractionation was evident in May (<90-µm fraction separated from the others), in August (<0.8-µm fraction separated from the others) and November (all three separated), while there was no effect of size fractionation in March.

The community composition, visualized as a heat map in Figure 6, shows a general increase in Gammaproteobacteria in all the experiments and all fractions. At the genus level, we observed differences according both to season and size fraction. In March, *Colwellia*, *Balneatrix* and *Oleispira* became abundant in all fractions and *Moritella*, *Psychromonas* and *Marinomonas* in the two larger fractions. *Thaumarchaeota*, which was abundant at the start, was almost absent at the end of the experiment. A similar trend was seen in the May experiment, with two Gammaproteobacteria genera, *Colwellia* and *SAR92_clade*, increasing in abundance while the other initially abundant genera decreased (*JL-ETNP-Y6*, *Balneatrix* and *SAR86_clade*). In May, we also observed an increase in Flavobacteria, predominantly in the two larger fractions (T_0_: 11.1%, <0.8 µm: 7.9%, <10 µm: 21.0% and <90 µm: 53.4%). *Polaribacter* increased in all the fractions and *NS9-marine-group* in the <90-µm fraction. In the <90-µm fraction, we detected the highest increase in Alphaproteobacteria, mainly due to a substantial increase of *Sulfitobacter* (T_0_: 1.3%, <0.8 µm: 2.3%, <10 µm: 3.3% and <90 µm: 14.6%). We observed a similar development of Gammaproteobacteria in May and August, with an increase of *Colwellia*, *SAR92_clade* and *Balneatrix* and a decrease of *JL-ETNP-Y6* and *SAR86_clade*. Additionally, there was an increase of some genera, including *Dasania* and *Glaciecola*, that were only detected in the larger fractions. In the August experiment, we generally observed higher abundances of Flavobacteria and Alphaproteobacteria, with the highest increase measured in the <0.8-µm fraction for *NS5_marine_group*, *Polaribacter*, *Roseobacter* and *Sulfitobacter*. In November, the T_0_ community composition was similar to the March community, with high abundances of *Thaumarchaeota* (25.2%). The fraction of *Thaumarchaeota* increased towards the end of the experiment in the two smallest size fractions, whereas it decreased in the <90-µm fraction. In the largest fraction, we observed an increase of the *Thermoplasmata* Marine Group II Archaea. Additionally, both Alphaproteobacteria and Flavobacteria were found in higher abundances than in March. The Gammaproteobacteria genera with the highest increase in abundance were *Colwellia* and *Balneatrix*.

To evaluate changes in the metabolic potential, we used the community composition (16S rRNA gene data) at the start and end of the experiments to predict and quantify the metabolic pathways in terms of KEGG Orthology (KO) abundance using the Tax4Fun package in R (Figure A3). In March and May, almost all the metabolic pathways had an increase in genes associated to them at the end of the experiment. The opposite was evident for the experiments in August and November when the number of KOs assigned to the most abundant metabolic processes decreased by the end of the experiment. The pathways with the highest increase were the two-component system, biosynthesis of secondary metabolites, ABC transporters, biofilm formation and quorum sensing in the May experiment. The strongest decrease was seen in August for biofilm formation, the two-component system and biosynthesis of the secondary metabolites.

## 4. Discussion

The five grazer exclusion experiments were performed at time points representing the typical succession within the Arctic marine ecosystem (Figure 2 and Figure A1) [6,8,10,39]. In short, the late-winter period with no light, low biological activity and biomass, with a microbial community dominated by dinoflagellates, set the scene for the January experiment. In March, the community was still influenced by winter processes and low biomass dominated by dinoflagellates but with light just starting to return. During the course of the experiment, a slight increase in phytoplankton biomass was evident, indicating the transition from winter to spring. In May, with 24 h of sunlight, the biomass, which was dominated by phytoplankton, was significantly higher at the start of the experiment and accumulated over the 6 days of incubation. Typical spring bloom diatom species were the main primary producers (Figure A4) [39]. In August, the initial biomass was as high as in May but decreased during the experiment. The protist community changed from diatom-dominated to a more complex community with *Phaeocystis*, cryptophytes, nanoflagellates (autotroph and heterotroph) and a few species of diatoms. At the start of the November experiment, sunlight was almost completely absent, and the protist community composition comprised fewer autotrophic organisms with dinoflagellates, with *Phaeocystis* being the most important, and the biomass was highly reduced compared to the two summer experiments.

The aim of the experiments reported here was to investigate how the environmental conditions described above and links within the microbial food web affected the bacterial community with respect to the net bacterial growth, bacterial production and bacterial community composition. We hypothesized two potential outcomes. Firstly, that different size fractions would contain different bacterial communities with activity and composition changes over time (I), or secondly, that all fractions would remain similar throughout the experiment (II). The first scenario (I) could be the result of either the co-association of certain bacteria with other organisms that increase in abundance (Ia) or the targeted grazing of bacteria (Ib). The second scenario (II) also had two possible underlying causes: certain bacterial growth was associated with the dissolved substrates passing through all the filter sizes (IIa) or nonspecies specific grazing, reducing the overall abundance without changing the community composition (IIb).

Based on the results, we can identify these different scenarios and the drivers with the largest effect on the bacterial community at the five different time points, all characteristic for seasonal differences in the Arctic marine ecosystem. In January, the net growth changed insignificantly in all three size fractions. The microbial abundance was low, but we observed a substantial increase in bacterial production (BP) in the <10- and <90-µm fractions. Since enhanced BP was not coupled to the growth of any other organism groups, the higher productivity was likely based on organic substrates (particulate organic matter) larger than <0.8 µm. Such microaggregates have been described to be hotspots of microbial activity [40]. Alternatively, the experimental set-up was biased towards larger bacteria, which were excluded from the <0.8-µm fraction. We did indeed see a reduction in bacterial abundance in the <0.8-µm fraction. Stocker [41] suggested that large bacteria represent one evolutionary line of metabolically more flexible and active bacteria (r-strategists), as opposed to small, nonmotile species/strains with streamlined genomes unable to respond quickly to altered conditions. The high BP observed in the two larger fractions could be an expression of large (r-strategic) bacteria becoming active quite rapidly, whereas the low activity measured in the <0.8-µm fraction could have been caused by the experimental set-up with too-short a time for the small k-strategists to become active.

For the January experiment, the community composition data was lacking, but since both the in situ surface community composition [7] and the outcome of the experiments (Figure 3) were comparable in January and March, we assumed that the ecosystem did not change much during this period.

A higher BP in the two larger size fractions might be a result of the increase of specific Gammaproteobacteria not present in the <0.8-µm fraction, namely *Moritella*, *Psychromonas* and *Marinomonas* (Figure 6). Such an increase of certain genera limited to the larger fractions indicated a co-association with the growth of specific diatoms (scenario: Ia), which, due to their size, were restricted to the larger fractions. Such co-association and reciprocal growth have previously been shown between *Moritella* and *Marinomonas* and diatom species blooming in the early spring [42]. Other Gammaproteobacteria, including *SAR92* and *SAR86* clades, have also been associated with early spring blooming conditions [43,44]. These co-associations suggest beneficial relations and directed interactions and are supported by our functional predictions (Figure A4). We identified an increased expression of genes associated with quorum sensing, flagellar assembly and biofilm formation during the March experiment. *Moritella*, *Psychromonas* and *Marinomonas* have previously been associated with particles during phytoplankton blooming and increased more than any other Gammaproteobacteria in the fractions also containing phytoplankton [45,46,47,48,49]. The genome analyses revealed the presence of siderophore biosynthetic genes [45,46,47,48,49] necessary for iron acquisition and linked to eukaryote-associated lifestyles [50,51]. Especially during the early phytoplankton bloom stages, carbohydrates, amino acids and other secondary metabolites are produced and released by phytoplankton and serve as chemoattractants for beneficial bacteria that, in turn, provide substances like iron needed for further growth [12]. An increase of the typical early bloom stage Gammaproteobacteria species limited to the larger size fractions thus indicates that, already in March, an association between beneficial bacteria and diatoms might lay the foundation for later onsets of blooms in late April/May.

In May, the environment was characterized by the spring bloom. Larger phytoplankton were, as intended, only present and growing in the <90-µm fraction (Figure 1, Figure 2 and Figure A1), which was also the size fraction where the bacterial community composition changed the most during the experiment. The community composition was different between the three size fractions (Bray–Curtis-similarity: <90-µm fraction was 46% similar compared to the <10-µm and 37% to the <0.8-µm fraction), while the bacterial production was relatively similar. This suggests that the bacterial activity (i.e., net growth and BP) was controlled by the availability of the dissolved substrates (scenario: IIa), while the community composition was determined by the presence of phytoplankton (scenario: Ia). Such selection led to an overall decrease in diversity (Figure 5a), particularly in the <90-µm fraction. The main differences between the largest fractions included a higher abundance of Flavobacteria (*Polaribacter* and *NS9-marine-group*) and Alphaproteobacteria, mainly the genus *Sulfitobacte* (Figure 6). *Sulfitobacter* is often associated with diatoms [52,53] such as *Chaetoceros*, which was highly abundant in the <90-µm fraction. The concurrent increase in the bacteria (*Sulfitobacter*) and the diatom (*Chaetoceros*) genera underpinned that this was a beneficial association [54,55,56]. The first sign of grazing activity, as indicated by an increased HNF abundance in the <90-µm fraction, was also seen during the May experiment. However, the increase in HNF did not lead to a decrease in abundance of the other microorganisms that we measured (Figure 2), showing that, at this point of the bloom succession, the growth of the prey (bacteria and picophytoplankton) exceeded the grazing loss.

The experiments in August showed the overall most clear effects of heterotrophic grazing in the two larger fractions, where the *Synechococcus* and bacterial abundances decreased markedly. Comparable reductions of the prokaryotic biomass in the <10- and the <90-µm fractions indicated that HNFs were the dominant bacterial grazers in August, in-line with previous studies identifying HNF (<5 µm) as the main top-down controller of bacterial abundance in the Arctic [57,58,59]. Furthermore, and similar to what Vaqué et al. observed [60], we only found a correlation between the abundance of HNF and HNA bacteria (R^2^ = 0.4580, *p* = 0.0453) and no significant correlation of HNF with the total bacteria abundance (Figure A5). This correlation was only apparent in the <10-µm fraction, which indicated that the grazing pressure in the larger fraction (<90 µm) by HNF was less strong, possibly due to the grazing pressure on HNF by microzooplankton [60,61,62].

The weak, yet positive, correlation between HNF and HNA bacterial abundance pointed to selective grazing of the active bacteria. The active bacteria, estimated as HNA abundance, correlated significantly with the relative abundance of Flavobacteria, including *Polaribacter*, *Ulvibacter*, *NS9-marine-group* and the Alphaproteobacteria *Sulfitobacter* (Table A1). A high abundance of these typically organic matter-degrading Flavobacteria in the <0.8-µm fraction further suggests that this group thrives in a post-bloom situation and that they are the main HNF prey. The observation that the community diversity was higher in the larger size fractions supports this interpretation. If the two most abundant groups found in the <0.8-µm fraction (*NS5-marine group* and *Polaribacter*) were removed by grazers in the larger size fractions, the groups otherwise outcompeted could grow, and the diversity increased accordingly. Our results and other studies [7,63] showed a general trend towards higher diversity in the Arctic winter. Selective grazing of the most dominant species and active forms during post-bloom conditions may be an important factor for explaining this shift in diversity.

The conditions in November were characteristic for early winter, with the reduced abundances and activity of most microorganisms. The BP and specific BP were low in all the fractions, indicating that resources were exhausted. Thus, neither substrates or co-associations with other organisms influenced the bacterial community composition, and there was no obvious differences between the size fractions. The initial bacterial community was similar to the one observed in March, with high abundances of *Thaumarchaeota*. However, in contrast to March, *Thaumarchaeota* remained stable or even increased in abundance over time. *Thaumarchaeota* are chemoautotrophic archaea and important for the nitrogen cycle oxidizing ammonia to nitrite, which is further oxidized to nitrate by *Nitrospinae* [7,64]. This increase in archaea indicates the return to a less productive Arctic winter season.

## 5. Conclusions

Using an experimental set-up with different size fractions allowed us to link specific bacteria to other groups in the microbial food web. These internal microbial interactions may be more central in shaping the diversity of the system than anticipated. We highlighted that the early associations of specific bacteria and diatoms in March might be essential for later phytoplankton growth in May. When the interactions switched from beneficial to opportunistic in August, grazing also played an important factor in shaping the community. This is summarized in a graphical abstract showing the specific changes in the Arctic microbial food web over the year and how the bacterial communities are affected by different drivers. We used bacterial community data coupled with measurements of the abundance and activity in different fractions to predict whether there were strong (positive or negative) interactions in the microbial food web. Detecting beneficial bacteria–phytoplankton associations can help predicting the overall growth benefits and, thereby, productivity in the microbial food web. Similarly, recognizing negative bacteria–phytoplankton associations may enable us to predict when the microbial food web is in a more recycling state, preventing carbon flow to larger organisms with less overall productivity. Identifying these interactions should be a focus for future studies in order to better understand the microbial component of the Arctic ecosystem and how different interactions affect community composition changes.

## Figures and Tables

**Figure 1 microorganisms-09-02378-f001:**
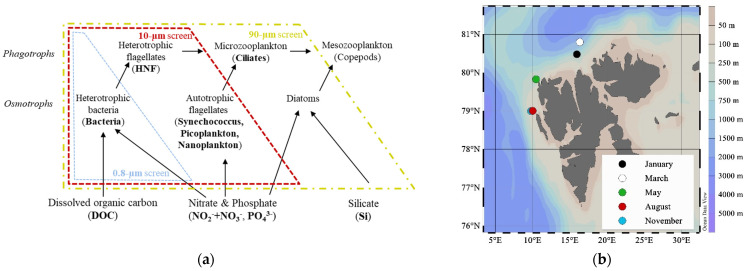
(**a**) Conceptual model of the microbial food web and the experimental manipulations by excluding different groups of phagotrophs (grazers) and osmotrophs (heterotrophic and autotrophic) using screens of different sizes (based on the “minimum” microbial food web model from Thingstad et al. 2007 [15]). Nutrients and organisms that were measured in this study are marked in bold and were placed in this model according to their size class and function. Frames of dashed lines in different colors indicate which organisms are included in the different size fractions. Size fractionation had no effects on the nutrient concentrations at the start of the experiment. (**b**) Location of stations NW of Svalbard sampled at different months in 2014 are indicated by the colors.

**Figure 2 microorganisms-09-02378-f002:**
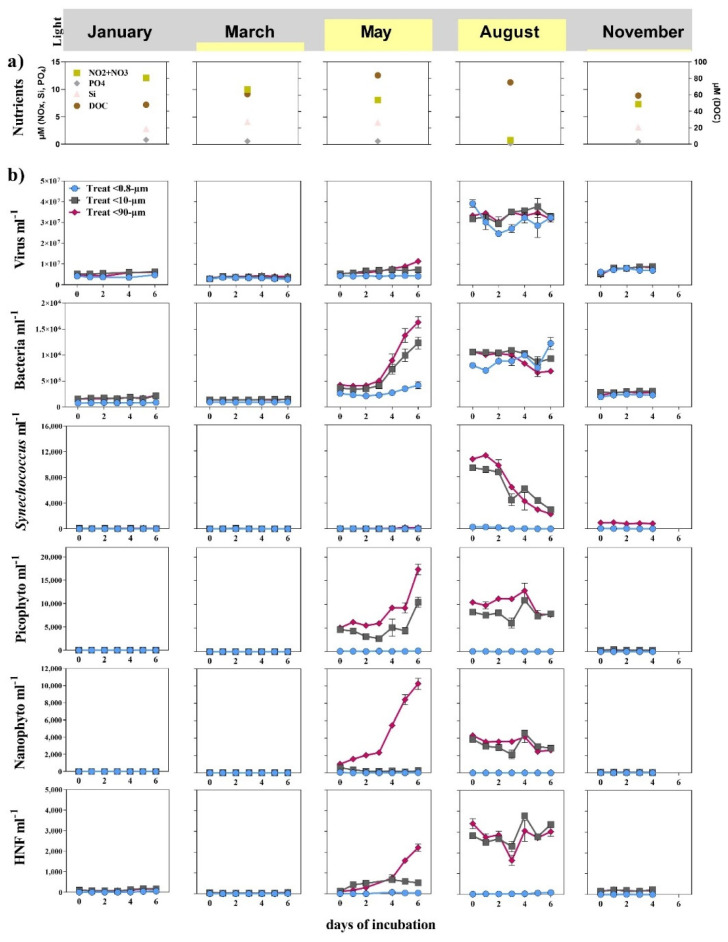
Different starting conditions for the five grazer exclusion experiments indicated in the yellow bars as a proxy of the daily light availability and (**a**) nutrient (NOx = nitrate and nitrite, Si = silicate and PO_4_ = phosphate) and dissolved organic carbon (DOC) concentrations (right y-axis). (**b**) Abundance (cells mL^−1^) of the microbial organisms measured during the incubations, including (from top to bottom) virus, bacteria, *Synechococcus*, picoeukaryotes, nanophytoplankton and HNF shown as cells or virus particles mL^−1^. The data is presented for three different size classes based on experimental manipulation carried out during five different months between 79° N and 81° N northwest of Svalbard.

**Figure 3 microorganisms-09-02378-f003:**
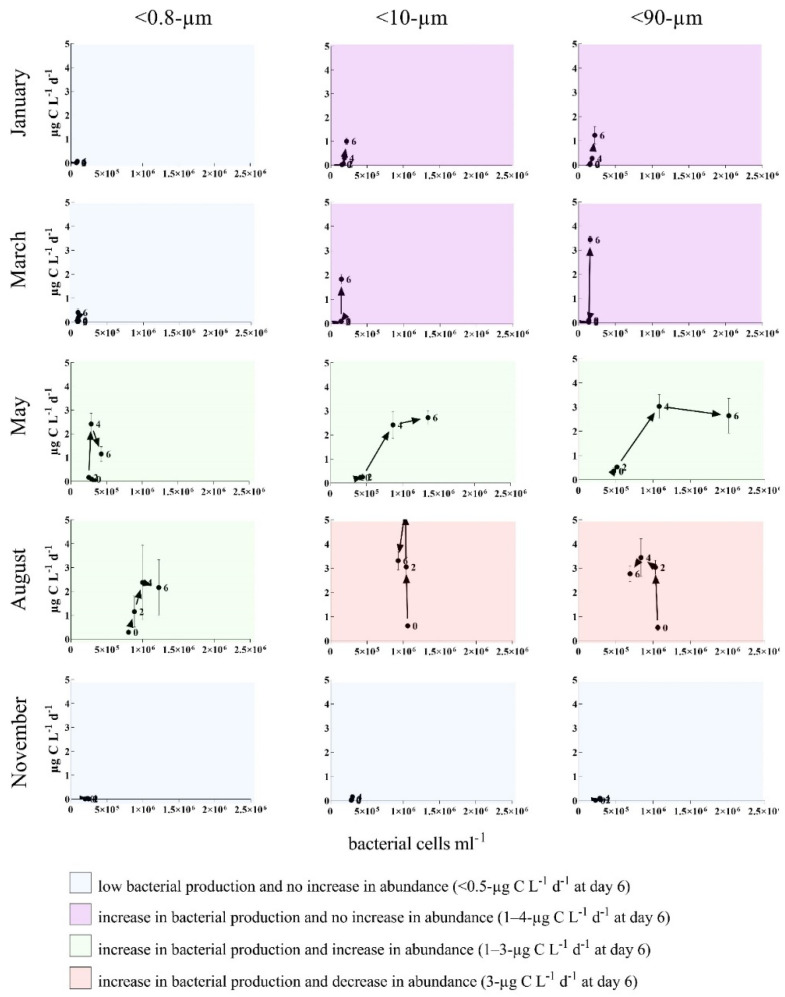
Plots of bacterial production (BP) shown in relation to bacterial abundance (BA) in the different fractions at 0, 2, 4 and 6 days of incubation. The different background colors indicate different groups based on a combination of BP and BA. Plots with a blue background indicate low BP and no increase in BA (<0.5-µg C L^−1^ d^−1^ at day 6). Purple indicates an increase in BP, but no increase in BA (1–4-µg C L^−1^ d^−1^ at day 6). Green visualizes experiments with an increase in BP and increase in BA (1–3-µg C L^−1^ d^−1^ at day 6), while red indicates an increase in BP and decrease in BA (3-µg C L^−1^ d^−1^ at day 6).

**Figure 4 microorganisms-09-02378-f004:**
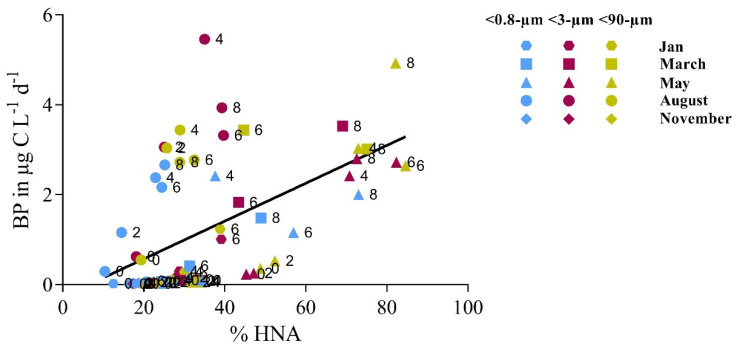
Bacterial production (BP) plotted against the percentage of high nucleic acid bacteria (% HNA) of the total bacteria counts. Numbers next to the data points indicate the time point of the incubation when BP was measured. Colors indicate the different fractions and symbols the different months when the experiments were performed (Spearman’s *r*: 0.63; Type II regression: *p* < 0.0001).

**Figure 5 microorganisms-09-02378-f005:**
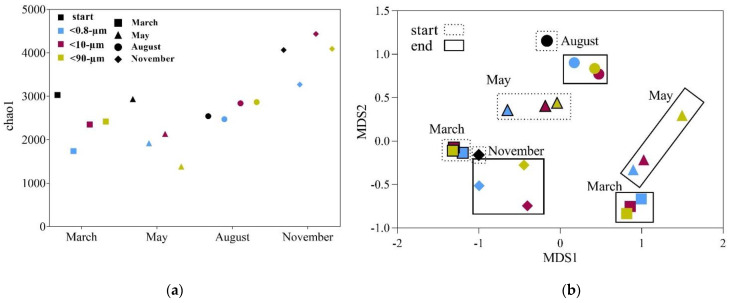
(**a**) Alpha diversity calculated for the estimator chao1 at the start (T_0_) and the end (T_end_) of the experiments for the three different fractions. (**b**) Multidimensional scaling (MDS) plot, illustrating the differences in community compositions between the different experiments and size fractions (<0.8-µm, <10-µm and <90-µm). Colors indicate the different fractions and symbols the different months when experiments were performed. Samples representing the start conditions are grouped within dashed boxes, while samples representing the T_end_ are grouped within solid boxes. The sample set taken for sequencing from the January experiments was incomplete, and the data is therefore not included.

**Figure 6 microorganisms-09-02378-f006:**
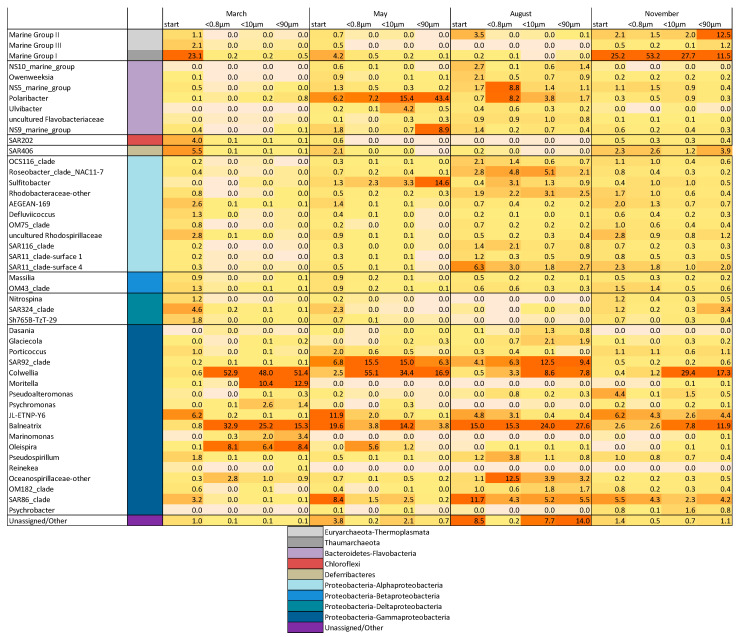
Heat map showing the changes in relative abundance of the highly represented (relative abundance >1%) bacterial taxa (genus level) based on 16S rRNA gene sequencing from the start to the end of the incubation for the different size fractions (<0.8-µm, <10-µm and <90-µm). Relative abundance is indicated as a percentage and illustrated by a shade of orange, with dark shades for high and light shades for low values. The phylum and class level, if relevant, are indicated via a color code on the lower right-hand side.

## Data Availability

All sequencing data is available at “The European Bioinformatics Institute” under study accession number PRJEB47254. (https://www.ebi.ac.uk/h, accessed on 27 October 2021).

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
