# Peer review of "How Microbial Food Web Interactions Shape the Arctic Ocean Bacterial Community Revealed by Size Fractionation Experiments"

_microorganisms, 2021, doi:10.3390/microorganisms9112378_

Round 1
Reviewer 1 Report
I have read all comments made by other reviewers and I also checked for changes requested in my own review. Since the authors have changed the manuscript accordingly, I think it is now a better version that can be published without further changes.
Author Response
We thank reviewer 1 for taking the time to evaluate the changes made in our resubmitted manuscript and are pleased to hear that these changes improved our manuscript.
Reviewer 2 Report
microorganisms-1463708
How microbial food web interactions shape the Arctic Ocean bacterial community revealed by size fractionation experiments
General comments:
The present study provides experimental results through manipulation of microbial food web using size fractionation, uncovering microbial interactions during an annual cycle, at the Arctic Ocean. With these results the authors show how the Arctic microbial food web is changing over the year and how the bacterial communities are affected by different biological drivers. It is extremely important to study and monitor these ecosystems as they are considered one of the most unstable ecologically fragile of earth, subject to changes related to climate variability and anthropogenic impacts.
Overall, the manuscript is well written and structured. This is an improved version of the previously submitted, and the authors carefully tried to follow the reviewers’ suggestions and answer the questions posed to clarify some aspects. It was a huge improve and I believe the manuscript is almost ready for publication. However, I still have a few suggestions that are described below in the specific comments.
Specific comments:
Introduction
Page 1, Line 36: I suggest changing to: “…and phagotrophic grazers.”
Page 2, Line 47: I would add a comma in this sentence: “This period, with increased phytoplankton activity and productivity, is relatively short in the Arctic [10,11].”
Materials and Methods
Page 4, Line 152: I would put the name of the authors before the citation number [28].
Page 4, Line 172: Delete the comma after the citation: “…following the protocol of Zubkov et al. [22].”
Page 4, Line 174: I believe you did not cite in the text reference number 30.
Page 5, Line 201: Correct to: “DNA and RNA were co-extracted…”
Page 5, Line 212: When you cite Wilson et al., remove the comma and the year, and put reference number.
Discussion
Page 14, Line 421: Correct to: “…fewer autotrophic organisms…”
Page 14, Line 430: Correct to: “…will remain similar throughout the experiment (II).”
Page 14, Line 446: Remove the year: “Stocker [42] suggests…”
Page 15, Line 504: When you cite Vaqué et al. remove the year and put only the reference number.
References
In the reference number 32 you have the names of the authors repeated, correct it.
Author Response
We thank reviewer 2 for the positive review and for further comments. We included all suggested improvements (listed under specific comments) accordingly. Furthermore, after several rounds of careful proof reading, we identified a few additional formal inconsistencies that we fixed in the new re-submitted manuscript. All changes are highlighted via track-changes.
This manuscript is a resubmission of an earlier submission. The following is a list of the peer review reports and author responses from that submission.
Round 1
Reviewer 1 Report
Journal: Microorganisms
Manuscript ID: microorganisms-1383618
Type of manuscript: Article
Title: How microbial food web interactions shape the Arctic Ocean bacterial community revealed by size fractionation experiments
Authors: Oliver Müller *, Lena Seuthe, Bernadette Pree, Gunnar Bratbak, Aud
Larsen, Maria Lund Paulsen
Submitted to section: Environmental Microbiology,
Major Comments
This manuscript showed data want to understand how community changes at the base of the food web affect interactions with other microorganisms and how this might cascade up to larger key predators, such as micro- and mesozooplankton, and they first have to unravel what drivers of bacterial community composition are. They hypothesize that bacterial growth is highest during the productive time of year, when nutrients are still high, and that phytoplankton abundance and composition deter mines bacterial community structure. During post-bloom phases when regenerated production dominates, we expect grazing pressure to be highest, and hypothesize top-down control to be an important driver of bacterial community structure.
After reviewed this manuscript, I had some concerned about the structure of the text. Firstly, it is clear to show temporal variations of Synechococcus and picoeukaryotic abundance in Figure 2, however, the authors did not show the detail data of picophytoplankton in model of Figure 1b, I'm totally confused here. Secondly, the authors showed the data of BP and Chl a in Figure 3 and 4, however, they did not show the material and methods about BP and Chl a in Methods section. To me, I think this paper has not been well characterized as of yet, so I strongly encourage the authors to reanalyze their data and make the appropriate modifications to the manuscript. This manuscript needs to be addressed and the results and discussion rewritten to focus on the new analysis. With the above points in mind, at present, I cannot recommend its publication in Microorganisms journal.
Specific Comments
Lines 22-23. I'm confused here, “Grazing pressure from bacterivorous microzoo plankton had an overall limited effect on bacterial community composition in which size? This the authors mean the major grazers on bacteria is microzooplankton? And how about the grazing effects of nanoflagellates on bacteria in this study? The authors did not show any results on this topic? Please reword this sentence.
Lines 37-39. “One of the key processes is the cycling of carbon, including the incorporation of inorganic and organic carbon into new biomass and growth and the transport of this biomass into higher trophic levels via grazing, also known as the microbial loop”… Please show the role of bacteria, nanoflagellates, ciliates and viruses in microbial loop clearly.
Line 58, “small phytoplankton” What kind of small phytoplankton in sized? < 2 μm or < 20 μm?
Lines 65-66. “Environmental changes occurring in the Arctic will affect the organism at the base of the food web and therefore have the potential to disturb the entire ecosystem function”. What kinds of environmental factors change in the Arctic will affect the microbial communities? Temperature, nutrients or light? Please show the previous studies in the Arctic about microbial communities in this paragraph.
In Introduction section, the authors should arrange the previous studies in other study areas to show the relative importance of top-down and bottom up control on bacterial communities, and these parts are need to be introduced earlier in the text.
In “Materials and Methods” section, the authors showed the methods of BP, Chl a, bacterial growth rate…, because you showed the data in Figures in the text.
Could you shorten your results section, it is too long to read clearly. In addition, I cannot find any analysis to show the control factors of bacterial and picophytoplankton and nanoflagellate abundance in Figure 2c in the Results section. Please focus on this topic more deeply.
Please remove the lines 235-250 to “Materials and Methods”.
At present, I cannot give any specific comments about Discussion section, this manuscript needs to be addressed and the results and discussion rewritten to focus on the new analysis.
Reviewer 2 Report
The authors report the community shifts in marine water samples taken at five different time points throughout the year in the Arctic Ocean upon size fractionation and incubation. To this aim the authors analysed the organism count, the bacterial activity by the means of bacterial productivity and cell count, the influence of high nucleic acid bacteria, the diversity of the 16S rRNA, the relative abundances of specific bacterial taxa bacterial and they estimate the functional capabilities of the found bacteria. The aim of the study was to gain better understanding of grazing effects on the bacterial community composition to understand transport of organic matter.
In the abstract, I find some redundancies that are not necessary and might just me removed. The same is true to some extend in the introduction that gives a lot of information on the effects of light intensity. I would also suggest mentioning and explaining some of the topics that will be discussed, such as HNF, bacterial production, regenerated production, verbalise the effects of grazing and top-down control. This does not have to be long but give the reader an intro in what will be talked about and hints on where to read further.
I am missing relevant information from the Methods section, regarding determination of bacterial production, microscopic counting of microplankton, nutrients and light plotted in Fig 1, chlorophyll a determination. This information will have to be contributed for publication.
I do also have some troubles with the results section. The way the data are presented in the figures makes it hard to see the relevant information, also the captions are most often not complete. Especially the different x-axes make it a big effort to understand the data. The text often does not help as it is very descriptive and does not guide the attention of the reader. Also, there are some interesting results in the supplement that are not mentioned in the results section. Going on to the discussion section, there are few references to the discussed results and figures, so that the reader must guess what is being talked about, for example lines 363 to 410. One solution might be to combine results and discussion but this of course up to the authors.
From reviewing the made analyses and the discussion as far as it was feasible with the lack of information introduced by the Methods section and the suboptimal figures, I however liked the question and how it was approached. Also, the discussion is very pleasant to read and seems to address all necessary aspects, making the lack of cohesiveness in the other chapters even more noticeable.
Therefore, I would like to suggest that the authors address my remarks before I continue with the review as I do not feel enabled to do it properly now.
Specific comments:
11: It seems the authors are using the terms ‘Arctic’ and ‘Arctic ocean’ throughout the manuscript. As they are more experienced with the usage of these terms, I would just ask to see through the manuscript if and where changes may be advisable.
13: Suggestion to change to “…the major seasonal transitions in microbial communities, dominated by chemolithotrophic Archaea in winter and phytoplankton associated with fast-growing, carbon-degrading bacteria in summer.
15: Replace “While light availability overall 15 is driving the seasonal changes” as it is redundant. Suggestion: “During these seasonal changes”
17: Phytoplankton-bacteria associations were already mentioned in the last sentence as overlaying effect.
20: Please settle on one way to refer to proteobacteria, either with or without greek letters but not mixed as is now.
35: delete “all”
43: either add or loose a comma
46: is nitrogen considered a mineral nutrient?
47, 48, 92, 95,112,164, 273: Please introduce abbreviations (or do not use them where they are not necessary)
47: Figure 1b is referenced before Figure 1a. It seems to be easy enough to switch them.
59: Move “will” to “spring will rise”.
61-64: Please provide references.
67: add “of” to “inflow of Atlantic water”
69: Put “Water” in lowercase.
74: Suggestion: “…we first have to unravel the drivers of…”
83: Replace “drivers of the bacterial community” by “processes”, otherwise the phrasing becomes a bit redundant
89: Please settle on a way to write the value and the unit (either with or without space) and stick to it in this whole chapter.
94-97: Is this important or can it go to streamline the information a bit?
154: The Greengenes database has been discontinued as of 2013. I do not think that redoing the analyses will change the overall results here but please keep it in mind for future analyses.
160: SILVA is not a functional but a rRNA database, please repair this sentence.
186: Italicise Synechococcus.
194: add comma after “May”
201: There is a superscript 1 missing.
235-250: I would have liked to read that paragraph very early in the results chapter. Please move it somewhere appropriate.
238: This is an excellent idea (the term excellent here referring to all meanings imaginable as it also includes “really confusing, wrong and unnecessary”). If you mean prokaryotes, then why not say prokaryotes? Please fix.
251: I did not find in the Methods section hoe the bacterial production was measured.
331: Typo.
359: Too many capitals.
Figure 1: Please severely upgrade you figure caption. What do we see in A? References and abbreviations for B.
Figure2: A and B) where do the underlying values come from? Add measurements to Methods section. Please also give a bit context in captions. C) I am not very happy with the changing y-axes, because it makes processing so much harder for the reader as necessary. Please find a way to visualise your data in way that helps the reader. I hope this will also address the issue of sometimes giving values on the ticks and sometimes not (I see what you did there, but it did not have the effect you hoped for). In the caption you mention that the samples were taken at 79°N, but in Figure 1a it says differently. Please fix. Again, italicise genus names.
Figure 3: Firstly, I wonder why the chlorophyll a results are included in this figure as they are not really discussed in the text, especially not in the context of the other values. Removing it from this figure might open a possibility to make it more accessible to the reader. Again, I find the different scales (x and y) disadvantageous. However, I like the color-coding idea, but maybe include a legend so it is easier to grasp.
Figure 4: Why are there several identical dots. Do they derive from the different incubation days? Please clarify.
Figure 5: Might be good to use the same style for A and B, especially the color-coding could help.
Figure A1: caption says that the figure shows the initial abundance of microbial organisms as weel as after six days of incubation. Does this make sense?
Figure A2: X-axis caption is wrong. Again, why use different y-axis.
Figure A4: Please do explain Methods.
Figure A5: You will have guessed by now: why change the y-axes.
Reviewer 3 Report
The conclusion part is well structured, but the authors should highlight their original results briefly rather than rely on general statements.
We detected a number of typographical errors (lines 425, 426; experiemnts, abudnances), so the authors should review the manuscript once again.
Reviewer 4 Report
microorganisms-1383618
How microbial food web interactions shape the Arctic Ocean bacterial community revealed by size fractionation experiments
General comments:
The present study provides experimental results through manipulation of microbial food web using size fractionation, uncovering microbial interactions during an annual cycle, at the Arctic Ocean. With these results the authors show how the Arctic microbial food web is changing over the year and how the bacterial communities are affected by different biological drivers. It is extremely important to study and monitor these ecosystems as they are considered one of the most unstable ecologically fragile of earth, subject to changes related to climate variability and anthropogenic impacts.
Overall, the manuscript is well written and structured, therefore I recommend its publication. However, I still have a few suggestions and comments that will help improving the manuscript, that are described below in the specific comments.
Specific comments:
Abstract
Page 1, Line 18: I suggest changing to: “Here, we uncover microbial interactions…”
Introduction
Page 2, Line 59: I believe you meant: “…and early spring, with the retreating ice edge…”
Page 2, Line 65: Correct to: “…will affect the organisms at the base…”
Page 2, Line 67: I suggest changing to: “…to the increased Atlantic water inflow…”
Page 2, Line 74: I suggest changing to: “…we first have to unravel what are the drivers affecting the bacterial community composition. Therefore, we aimed…”
Page 2, Line 82: I suggest changing to “We aimed at identifying…”
Materials and Methods
Page 2, Line 96: Regarding the description of the transects performed during the sampling period, I believe you should refer here only the months January and March, as you already mentioned August in the sentence before.
Page 3, Line 98: Did you sampled water from 20m depth only during May and August? Later in line 105, you mention that you used surface water samples in the experiments. Can you clarify this?
Page 3, Line 101: You should mention in the caption of Figure 1, the information of a) and not only b).
Page 3, Line 117: When you say: “Samples for microbial community composition were taken at the start and end of the experiments in March, May, August and November.”, does it mean that you did not take in January? You should mention it here and explain why.
Page 3, Lines 129 and 134: You should put in superscript “μL min-1”.
Page 3, Line 130: Correct to: “…green fluorescence (BL1) vs. red fluorescence (BL3).”
Page 4, Lines 163 and 164: You should define KEGG and KO.
Page 4, Line 168: I believe you should give more information about the statistical analysis; you only refer the analysis you used, you should also say in what data you applied them and explain the purpose of it. In fact, later in the results you mention it, but I believe it should be in this section.
Results
Page 5, Line 191: You should change to: “…in May and August (Figures 2c and A1).
Page 5, Line 192: You mention here that all larger microorganisms were not detected in the <0.8 µm fraction at any time point of the experiments, but in Figure 2c it is possible to see a slightly increase of HNF density in the January experiments, and of Synechococcus in March and May. Can you clarify this? Also, you should write in italic Synechococcus, check all the paper, and correct it.
Page 5, Line 202: I suggest changing to: “Growth initially presented highest values in the larger size fractions (10 and 90 μm) in May…”
Page 5, Line 227: Do you mean: “…after six days (4000 to 3000 cells mL-1).”
Page 10, Lines 307-310: As I mentioned before, this sentence should be placed in the method’s statistical analysis. Also, the sentence in lines 351 to 354 in page 11.
Discussion
Page 12, Line 390: Correct to: “…for two underlying causes: certain bacterial…”
Page 12, Line 415: Correct to: “…genera that we did not observe…”
Page 13, Line 422: You should put the numbers of the references instead of the names: “…associated groups [33,34], including SAR92…” Also, in the reference list you should put the names of the authors of the reference 33.
Page 13, Line 425: Correct to: “…March experiments increased abundances of genes…”
Page 13, Line 454: I suggest changing to: “No direct decrease of any other group was observed (Figure 2), however, it indicates that at this point…”
Page 13, Line 463: You should also put the number of the reference here “Similar to Vaqué et al.,”